# What Is It like to Be a Brain Organoid? Phenomenal Consciousness in a Biological Neural Network

**DOI:** 10.3390/e25091328

**Published:** 2023-09-13

**Authors:** Ivanna Montoya, Daniel Montoya

**Affiliations:** Department of Psychology, Fayetteville State University, Fayetteville, NC 28301, USA; imontoya@broncos.uncfsu.edu

**Keywords:** neural correlates of consciousness, biological neural network, phenomenal consciousness, artificial intelligence, information integration theory

## Abstract

It has been shown that three-dimensional self-assembled multicellular structures derived from human pluripotent stem cells show electrical activity similar to EEG. More recently, neurons were successfully embedded in digital game worlds. The biologically inspired neural network (BNN), expressing human cortical cells, was able to show internal modification and learn the task at hand (predicting the trajectory of a digital ball while moving a digital paddle). In other words, the system allowed to read motor information and write sensory data into cell cultures. In this article, we discuss Neural Correlates of Consciousness (NCC) theories, and their capacity to predict or even allow for consciousness in a BNN. We found that Information Integration Theory (IIT) is the only NCC that offers the possibility for a BNN to show consciousness, since the Φ value in the BNN is >0. In other words, the recording of real-time neural activity responding to environmental stimuli. IIT argues that any system capable of integrating information will have some degree of phenomenal consciousness. We argue that the pattern of activity appearing in the BNN, with increased density of sensory information leading to better performance, implies that the BNN could be conscious. This may have profound implications from a psychological, philosophical, and ethical perspective.

## 1. Introduction

The last few years have seen a surge of interest in figuring out the neural mechanisms that underlie consciousness, thanks to some fascinating technical developments. Work by Setia and R Muotri [1] showed that three-dimensional structures made from human pluripotent stem cells can show electrical activity similar to that of an EEG. These organoids can be generated from human pluripotent stem cells (hPSCs). During differentiation, brain organoids can be patterned to a specific brain region, such as the cortex. The organoids have been proposed as a viable model to test the effects of viruses or environmental toxins on human development, modeling genetic neurological disorders or as oncological models. The challenge has been developing strategies for three-dimensional electrophysiology. However, several recent developments have made possible electrophysiological recording in 3D using brain organoids [2].

Based on these technologies, the recent instantiation of neurons embedded in a digital game world has advanced the capacity to read motor information and write sensory data into cell cultures [3]. This report is the first to establish neural “wetware” for computational purposes. Pluripotent cells developed morphological characteristics of cortical neurons, with axonal and dendritic connections, showing electrochemical properties. The cells, instantiated in a simulated environment (*DishBrain*), exhibited spontaneous and evolving action potentials. The system allowed the recording of electrical activity from the neural culture while providing sensory information, a basic condition to establish its computational nature.

Moreover, these biologically inspired neural networks (BNN) were able to show internal modification and learn the task at hand (predicting the trajectory of a digital “ball” while moving a digital “paddle” to counteract it in a game of “Pong”). The study indicated that mice and human cortical cells showed evidence of learning compared with electrically inactive non-neural cells (HEK293T cells) and media-only controls.

Taken together, these experiments lead to some compelling inquiries at many levels. Certainly, the authors discuss the importance of their work in terms of technical achievement. After all, this is the first “*demonstration of a synthetic biological intelligence (SBI) system that learns over time in a systematic manner directed by input*”. The system also provides “*a theoretical framework for how intelligent behavior may arise*” by promoting the minimization of entropy.

However, one of the main questions relates to the real possibility that these biological neural networks show some form of phenomenal consciousness [4]. We start by asking how well our existing Neural Correlation of Consciousness (NCC) theories can predict the appearance of biological neural networks capable of learning in a virtual environment, somehow replicating the behavior of their body-instantiated counterparts.

In what follows, we will point out the implications that BNN’s development has for our understanding of consciousness and artificial intelligence. To accomplish that, we first present a brief description of our current state of NCC theories. For brevity’s sake, we can describe the implied theories only in the most superficial of terms. A more comprehensive description can be found in the attached bibliography.

## 2. Neural Correlation of Consciousness Theories

According to one definition [5], “*An NCC is a minimal neural system N such that there is a mapping from states of N to states of consciousness, where a given state of N is sufficient under conditions C, for the corresponding state of consciousness*”. The mapping between these two elements can be problematic in the case at hand and it is a two-pronged issue. We have on one side neural activity from a cluster of in vitro neurons recorded in close relationship with neural stimulation. The difficulty lies in what exactly can be considered a “*state of consciousness*” in an organoid. However, a useful approach would be to treat their neural activity (spike count) as producing something akin to a state of phenomenal consciousness.

Existing NCC theories understand the problem of consciousness arising from a whole brain, without defining a minimal neural system capable of sustaining consciousness. That could be understood as a limitation on how an in vitro cluster of neurons can be treated as a neural correlate of consciousness. With all these limitations in mind, we need to point out that, nevertheless, we have a group of neurons producing neural activity closely tracking environmental stimulation.

NCC theories assume that the entire brain is necessary and sufficient to produce consciousness. The question of interest then is which of its subcomponents are essential to produce a conscious experience. They also rely on the anatomy of a human brain without much possibility of addressing alternative systems either in vivo or in vitro.

In general, all NCCs provide a series of requirements and thresholds to grant consciousness to an agent, based on the central idea that consciousness may arise only from specific areas of the brain and the neural tissue associated with it. These theories thrive in their capacity to point out brain mechanisms that could sustain different degrees or forms of consciousness. In that sense, they possess an intrinsic heuristic value by offering a material grounding to the theory from which to make predictions. Progress in solving the mind–body problem may arrive from focusing on questions that are relatively straightforward instead of discussing philosophical arguments that may be too complex or abstract to puzzle out experimentally. In other words, a materialistic perspective on the easy problem of consciousness brings immediate rewards. The task at hand is the search for the neuronal correlate of consciousness and, ultimately, its causes. BNNs may be a useful tool to develop an explanation of how consciousness is instantiated in a neural system. However, we offer the caution that bioengineering developments should take theoretical models into consideration.

Simultaneously, it is crucial to highlight the significance of predictability within the context of scientific exploration. This pertains to the capacity to anticipate potential outcomes resulting from methodological and logical analyses of specific scientific concepts. This predictive aspect is a foundational cornerstone of scientific theories. In the case of Neural Correlation of Consciousness theories, which delve into intricate aspects of whole brain anatomy and advocate for an approach that starts with overarching principles to comprehend consciousness, their potential explanatory power remains uncertain. Succinctly, these theories may or may not have the foresight to suggest the feasibility of a revolutionary technological advancement—a brain-in-a-box scenario materializing through the transformation of stem cells into intricate biological neural networks.

The different proposals on how consciousness arises from cortical neuron interconnectivity are present in theories such as:Global Neuronal Workspace [6];Recurrent Processing Theory [7];Higher Order Theories;Information Integration Theory [8].

In what follows, we would like to summarily describe the most popular NCC theories available, looking at their capacity to predict the recent findings described by Kagan et al. [3].

### 2.1. Global Neuronal Workspace [6]

The central elements of the Global Neuronal Workspace theory (GNW) were proposed by Baar [6] as follows: “[9] *a state is conscious when and only when it (or its content) is present in the global neuronal workspace making the state (content) globally accessible to multiple systems including long-term memory, motor, evaluative, attentional and perceptual systems. This is a cognitive approach defending the idea that perceptual content becomes conscious only when they are widely shared by multiple brain systems or processors. It is the wide disponilility of this information on several processors at the same time what constitutes a conscious experience*” [6,10,11,12].

The global workspace can process long-term memory, perceptual information, and attentional systems while including inputs from value systems, motor plans, and verbal reports [13]. The theory was proposed originally at two levels. At one level, a network consisting of a set of parallel, distributed, and functionally specialized processors or modular subsystems, and, at another level, a computational space, a global workspace, consisting of a distributed set of cortical neurons [10]. As a result, it is closely tied to brain architecture.

Which workspace neurons constitute the workspace at a given time depends on the activity of those neurons given the subject’s current state. Moreover, the workspace is not a rigid structure, but a rapidly changing neural network that usually only contains a small number of cells. The model has been updated recently [14], by proposing a multilevel hierarchical system that comprises cognitive, conscious, and metacognitive levels, depending on a mechanism of synaptogenesis at a local and global level.

The GNW theory proposes that only cortical neurons can contribute to the phenomenon of consciousness. It also requires that several subsystems, such as long-term memory, attention, motor systems, etc., flow into this workspace. In the organoid system instituted on top of a computer chip, in our case at hand, cortical neurons are the main processing units, as required by GNW. Since the organic sensory neurons seem to predict or track the position of the ball, we could argue that a minimal attentional system is available. A simple motor system seems to respond, guiding the paddle to hit the ball.

However, the rest of the systems mentioned by GNW theory are definitively not available. We can point out, for example, the absence of a value system or a long-term memory system. Hence, we feel that this theory cannot predict or explain the behavior of the neuronal system described by Kagan et al. We may point out, nonetheless, that these are current limitations associated with the BNNs instantiation and not a theoretical drawback. Maybe future iterations of the technology may allow for the required systems suggested by GNW theory.

### 2.2. Recurrent Processing Theory [7]

This theory indicates that perceptual consciousness is a process that happens outside the workspace and focuses on recurring activity in sensory areas. The question from which Lamme departs is “What are we seeing?”. He then indicates that “different neural representations of a scene exist”. He also indicates that conscious percepts are often the result of “a *competition between several possible groupings and bindings*”. Using the example of the swift interaction between the visual and motor area (a fast forward sweep (FFS) of 200 ms in the cortex), he proceeds to explain how “*different aspects of objects and scenes are integrated into a coherent percept*”. According to Lemme, the properties of first-order neural representation explain consciousness. Recurrent processing occurs when sensory systems are highly interconnected and involve feedforward and feedback connections. The activation from some objects will travel all the way up to the motor and prefrontal areas, while others will only activate visual areas. According to Lamme [7,15], recurrent processing is necessary and sufficient for consciousness.

Lamme identifies four stages of normal visual processing:Stage 1: There is a first movement of superficial processing during the interaction between the visual and motor areas;Stage 2: A second stage of deep processing when the stimulus travels through the whole sensorimotor system and ends up in the prefrontal cortex, leading to behavioral changes;Stage 3: Given enough time for recurrent processing (RP) to occur, the third stage is the continuing superficial processing of a recurrent nature of the visual stimulus. “RP allows for dynamic interactions between areas that can grow ever more widespread as time after stimulus onset evolves”;Stage 4: A widespread RP occurs from low-level sensory information to high-level executive areas. This only happens when attention is engaged for sufficient time with the stimulus.

This last stage has been equated by some authors as the landing of the stimulus in the global workspace [16].

As previously noted for the GNW theory, Lamme’s account proves useful in understanding the brain mechanisms implicated in the close relationship between sensory and motor areas. It offers a simple process through which a percept can be transformed while submitted into higher-order areas. Nevertheless, the theory only applies to the human cortex (with some concessions to monkey’s brains since the visual data was heavily acquired from this animal model), but it does not open the possibility for any alternative system to be conscious.

Our organoid instantiated in a dish demonstrates a direct connection between sensory and motor areas, showing an example of what Lamme calls fast feedforward sweep (FFS), but that is how far the similarities go. There is no higher-order system in which the original percept is processed at a higher-order area. In addition, the central requirement of this theory, recurrent processing, is very limited in the internal connectivity of the organoid, since there are no cortical layers or complex cortical areas. Once again, this theory does not predict or explain Kagan et al.’s results.

### 2.3. Higher Order Theories

According to this set of theories, from higher-order perception theories [17] to higher-order thought theories [18,19], one is in a conscious state if and only if one can represent oneself as being in such a state. In other words, to be in a conscious state means one must be aware of it. There is a difference between conscious and unconscious mental states, the last ones being those states we are not aware of. The theorists propose a strong link between a mental state and the awareness of this mental state. In other words, one should be conscious to know that one is conscious. Beyond this circular definition, higher-order theories connect high-order representations with activity in the prefrontal cortex, which is taken to be the neural substrate of the required higher-order representations [20,21,22].

According to these theories, there should be a distinction between creature consciousness and mental state consciousness. This was conceptualized by Ned Block [4] as distinguishing between phenomenal consciousness (or phenomenology, for short) and access consciousness (or accessibility, for short). This means that a great swat of cognitive activity, being unconscious, is inaccessible to us.

Higher-order theories in general do not predict the occurrence of consciousness in such a simple system as an organoid. With their focus on self-awareness and higher-order thought, the theories disqualify themselves from explaining Kagan’s results. However, defining levels of consciousness is very useful, in the sense that phenomenal consciousness (P consciousness) could be invoked to understand or give context to this system. Granted, it would be a stretch to assume any form of self-awareness in a BNN, but the presence of a very basic form of creature consciousness cannot be ruled out.

### 2.4. Information Integration Theory 1.0 [8]

According to Tononi et al. [8] “*consciousness corresponds to the capacity of a system to integrate information*”. In his seminal paper, the author suggested that consciousness possesses two main properties: differentiation and integration. The first is defined by an extraordinary quantity of conscious experiences, and the second refers to the perceived unity of such experience. An example of differentiation is given by the perception of light and darkness in an image and the myriad of states in between offered to the visual system. An example of integration is the impossibility of separating shape from color. In other words, consciousness is a highly integrated phenomenon presenting itself as a globally unified field of experience. We can assume a very basic form of differentiation in the BNNs’ processing of information due to the existence of differentiated sensory–motor areas, along with a high level of integration since the BNN responds as a whole to the information.

Parallel to these main ideas, as with any substance, there are two other elements to consider, such as quantity and quality. The quantity of consciousness in a given system is expressed by “the Φ value of a complex of elements”. Quality, on the other side, is how much information is integrated across the system. Any physical system will have subjective phenomenal experience Φ to the extent that is capable of integrating information. Information in this context follows the original definition by Shannon [23] as a reduction of uncertainty. The higher the number of different states a system could have, the more information it carries, and the more any single state reduces uncertainty.

For Tononi, then, consciousness corresponds to the capacity to integrate information. He was not talking about correlation, which is a relationship between two or more similar objects that is reciprocal, parallel, or complementary, but correspondence, which is the exact form on opposite sides of a dividing line or plane, in this case, a Markov blanket. 

Of course, in order to be recognized as a NCC, the theory proposes, at least in its inception, an anatomical location to sustain conscious experience. According to Tononi, consciousness is generated by a distributed thalamocortical network, following Plum [24], invalidating the possibility that a single cortical area would be capable of consciousness. Current scientific communities assume that the physical substrate for the generation of consciousness is neural networks, with neural assemblies being activated and deactivated in parallel on countless brain structures [25]. Since IIT holds that a non-zero value for Φ implies that a neural system is conscious, with more consciousness present with greater values for Φ, this is the only theory that provides BNNs with the possibility of being conscious.

So far, because of its low application barrier, IIT is our best candidate to explain the BNN’s results. Since any system that is capable of integrating information shows some kind of consciousness, the IIT seems to be posed to predict the possibility that the organoids described by Kagan et al. show some form of phenomenal consciousness. In the following section, we briefly list the parallels between the experiment’s results and the features of IIT as proposed in its early version. The theory is currently in its version 4.0 [26].

## 3. Information Integration Theory Predicts the Results on Biological Neural Networks

“*A first claim is that the neural substrate of consciousness as we know it is a complex of high Φ that is capable of integrating a large amount of information—the main complex*”.[8]

From the first statement, “*Consciousness is the capacity of the system to integrate information*”, we sense the comparison is not banal. The BNN shows a dense interconnected neural network with spontaneous activity. As expressed by the authors, based on the sensorimotor recordings (Kagan et al.; Figure 3 in [3]), the system can carry high levels of integrated information. Moreover, these neurons can change their firing rate when the environment is unpredictable.

Tononi explains that “[9] *the quantity of consciousness available to a system can be measured as the Φ value of a complex of elements. Φ is the amount of causally effective information that can be integrated across the informational weakest link of a subset of elements*” ([8]). We can establish that, according to the reported results, information clearly flows through the BNN. A Φ value plainly different from 0 can be deduced from the firing rate of cortical cells differentiated from hiPSCs. This constitutes the “main complex” mentioned by Tononi.

Moreover, neural activity in the BNN changes in real time in response to environmental stimuli, and it shows performance improvement translated as “learning with feedback”. The authors also point out that increasing the density of sensory information input leads to increased performance, as translated into an increase in average rally length (hit-miss ratio). Based on this data, we believe that the BNN shows a high Φ value leading it to integrate a large amount of information in the form of sensory data (simulated environment), responding to external states and accommodating feedback (see Table 1).

“*A second claim of the theory is that the quality of consciousness is determined by the informational relationships within the main complex. Therefore, how a group of neurons contributes to consciousness is a function of its informational relationships inside the complex and not outside of it*”.[8]

A second statement from IIT indicates that the quality of the information traversing the system is determined by the informational relationship between elements of a complex. “A complex is a subset of elements with Φ > 0 that is not part of a subset of higher Φ” [8]. We can establish, roughly speaking, that the subdivision of sensory and two motor areas determines the internal informational relationship within the main complex, an elementary organization, of course. However, each of the areas is a subset of elements where Φ > 0, again, based on the firing rate and response to environmental data. We must remember that, according to Tononi, “a system is conscious when Φ > 0”. The recording electrodes were arranged in a way that would allow a topographically consistent place coding, while the motor activity was accumulated in real time to move the paddle. This distribution, we must point out, mimics the basic internal organization of a cognitive agent (see Table 1).

“*[9] Each particular conscious experience is specified by the value, at any given time, of the variables mediating informational interactions among the elements of a complex*”.[8]

In the experiment described by Kagan et al. [3], we are not referring to unconscious forms of learning such as classical conditioning or operant conditioning, but rather to an active form of learning that requires predicting environmental conditions (such as the ball’s position) and responding to them (such as the paddle’s position). Our hypothesis is that in each test, a sequence of a predictable trial in which the ball is released, and the neurons react to its position by moving the paddle, may constitute an experience for the system (see Table 1). We recognize that this is an uncomplicated system, but this should not distract us from the fact that its internal variables are mediating informational interactions among its elements.

### General Parts of IIT 4.0 Still Predict the Results on BNNs

In the most recent iteration of the theory [26] the focus moved to make it mathematically sound, turning it into a more abstract postulate. In fact, some general statements about the nature of consciousness are often of a higher-order nature. IIT 4.0 claims that experience exists and is of an irrefutable nature. We are talking about a physical experience when we refer to a system that can take and make a difference. From this axiom, five characteristics of consciousness are identified: “e*very experience is, for the experiencer, specific, unitary, definite, and structured*”. In a parallel comment [27], Albantakis indicates that the axioms of IIT should necessarily be seen as “*basic properties*”.

This version of the theory is still useful in our case because it is very specific and predictable. It also indicates that we need to define the smallest unit of consciousness, the neuron. Parallel to that, it requires a substrate of consciousness expressed in a set of units that can be observed and manipulated. All these attributes are readily available in the BNN system. However, in version 4.0, there is no reference to any particular brain region that could sustain consciousness.

## 4. Conclusions

In this paper, we tried to pose the important question if neurons instantiated in an in vitro system, connected to a virtual environment, are not only capable of exhibiting learning but also some form of consciousness. Based on the characteristics of the physiological response in the BNN, we can answer this question in the positive. The requirements laid out by Information Integration Theory 1.0 state that these neural organoids embedded in a game world exhibit signs of consciousness due to their capacity to integrate information. IIT offers a working definition of consciousness that can be applied in any system beyond the ones with thalamocortical pathways. The possibility that BNNs in a virtual world show some kind of P consciousness has profound ethical, engineering, and theoretical implications, but at the same time, this system may help us figure out, in the near future, what consciousness is.

Furthermore, these advances in BNNs represent the first implementation of a Turing machine in a biological system. We must remember that a Turing machine is a virtual device capable of reading, erasing, and writing symbols in an endless tape [28], a mechanism central to the processing of information. This proposal has been at the root of the strong AI hypothesis of the mind since early discussions [29]. Kagan’s experiment may be the first demonstration of the computational capabilities embedded in a biological system. Furthermore, the idea has been suggested that these experiences open the chance for an AI to be instantiated in a BNN. From a physiological, psychological, philosophical, and ethical perspective, many questions arise, not to mention other areas that can be potentially affected by BNN instantiation, such as medicine, gaming, transportation, or even military applications. From an ethical perspective, the issue concerning brain organoids has been discussed by Jeziorski et al. [30] and Lavazza et al. [31].

We do not know yet what consciousness actually is. However, we are now able to reproduce the substrate of consciousness in vitro, creating a new kind of sentient being. In that sense, we are in uncharted territory.

## Figures and Tables

**Table 1 entropy-25-01328-t001:** Shows that there are many similarities between the experiments with a BNN and the predictions made by IIT. We contrast some of the IIT’s fundamental requirements with the results obtained in vitro to suggest that this system might exhibit some degree of P consciousness.

	Integrated Information Theory 1.0	Biological Neural Network
**Activity**	Consciousness is the capacity of the system to integrate information	Dense interconnected networks with spontaneous activity
	The system carries integrated information Φ if information content in the whole is greater than the sum of the information in the parts	Learning is a property of the entire system
	A system is conscious when Φ > 0	Neural activity changes in real time in response to environmental stimuli
		Neurons adjust firing activity in sensory and motor areas in response to environmental unpredictability
		Increasing the density of sensory information input leads to increased performance
**Quality**	Quality of information is determined by informational relationship between elements of a complex. A complex is a subset of elements with Φ > 0 that is not part of a subset of higher Φ	BNNs divided into a sensory area and two motor areas with specialized activity.
		Mimics basic cortical organization in terms of sensory and motor areas
		Shows sensory–motor integration
		BNN shows performance improvement; “learning with feedback”
**Specificity**	Each particular conscious experience is specified by the value of variables mediating informational interactions among the elements of the complex	Sensory information delivered and motor information recorded in distinct trials.
		Each trail constitutes an “experience”
		Dynamics in electrophysiological activity display coherent connectivity

## Data Availability

Not applicable.

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
