# Peer review of "What Is It like to Be a Brain Organoid? Phenomenal Consciousness in a Biological Neural Network"

_entropy, 2023, doi:10.3390/e25091328_

Round 1

Reviewer 1 Report

Very interesting and professional discussion of the theory of neural correlates of consciousness (NCC). The authors question whether consciousness can be assumed to exist in a biologically inspired neural network (BNN). According to Information Integration Theory (IIT), any system capable of integrating information will have some degree of phenomenal consciousness. This question is immediately complicated - as it is likely that there are different forms of consciousness.

The question of this has a practical application - is a "consciousness meter" possible, and if so, on what principle should it work. This issue has been seriously considered by several groups, and specifically with regard to organelles. I would advise authors to use materials from these publications:

Koch C, How to Make a Consciousness Meter. Sci Am; 2017 Oct 19;317(5):28-33. doi: 10.1038/scientificamerican1117-28.  

Tsytsarev V., Methodological aspects of studying the mechanisms of consciousness. Behav Brain Res. 2022 Feb 15;419:113684. doi: 10.1016/j.bbr.2021.113684.

Jeziorski et al;  Brain organoids, consciousness, ethics and moral status; Review Semin Cell Dev Biol . 2023 Jul 30;144:97-102. doi: 10.1016/j.semcdb.2022.03.020. Epub 2022 Mar 23.

Lavazza A, Chinaia AA. Human cerebral organoids: the ethical stance of scientists. Stem Cell Res Ther. 2023 Apr 1;14(1):59. doi: 10.1186/s13287-023-03291-x.

The authors discuss the iteration theory of the Albantakis’ theory, which claims that physical experience exists and is irrefutable. However, it seems logical to use one of the recent works of this author, on this very issue:

Albantakis L. To be or to know? Information in the pristine present. Behav Brain Sci. 2022 Mar 23;45:e42. doi:10.1017/S0140525X21001977

 The material is presented accurately and logically, the manuscript will certainly be of interest to the scientific community. The above shortcomings are not of a fundamental nature and can be easily corrected. I will be happy to recommend the manuscript for publication after correcting the shortcomings outlined above.

Author Response

Response to Reviewer 1

Comment

The question of this has a practical application - is a "consciousness meter" possible, and if so, on what principle should it work. This issue has been seriously considered by several groups, and specifically with regard to organelles. I would advise authors to use materials from these publications:

Koch C, How to Make a Consciousness Meter. Sci Am; 2017 Oct 19;317(5):28-33. doi: 10.1038/scientificamerican1117-28.  

Tsytsarev V., Methodological aspects of studying the mechanisms of consciousness. Behav Brain Res. 2022 Feb 15;419:113684. doi: 10.1016/j.bbr.2021.113684.

Jeziorski et al;  Brain organoids, consciousness, ethics and moral status; Review Semin Cell Dev Biol . 2023 Jul 30;144:97-102. doi: 10.1016/j.semcdb.2022.03.020. Epub 2022 Mar 23.

Lavazza A, Chinaia AA. Human cerebral organoids: the ethical stance of scientists. Stem Cell Res Ther. 2023 Apr 1;14(1):59. doi: 10.1186/s13287-023-03291-x.

Answer

The existence of a “conscious meter” is an unavoidable next step when asking these kinds of questions in relationship to such special systems as brain organoids. While the suggested citations are very useful indeed, IIT and its capacity to allow a form of consciousness in an organoid remains the focus of our MS. The articles proposed do not deal specifically with IIT.

However, the suggested bibliography has been added to the MS in the appropriate places.

Comment 

The authors discuss the iteration theory of the Albantakis’ theory, which claims that physical experience exists and is irrefutable. However, it seems logical to use one of the recent works of this author, on this very issue:

Albantakis L. To be or to know? Information in the pristine present. Behav Brain Sci. 2022 Mar 23;45:e42. doi:10.1017/S0140525X21001977

Answer

The original paper from January 2023 from Albantakis et al (2023) was cited due to the in-depth presentation of IIT 4.0. The paper suggested by the reviewer is less specific for our discussion, since it is a comment on other article, but, nonetheless, it has been cited in the appropriate place.

Reviewer 2 Report

General Comments:

The authors aim to investigate whether a recent work from “Kagan, B. J., Kitchen, A. C., Tran, N. T., Habibollahi, F., Khajehnejad, M., Parker, B. J., ... & Friston, K. J. (2022). In vitro neurons learn and exhibit sentience when embodied in a simulated game-world. Neuron110(23), 3952-3969.” Is equipped with some form of consciousness apart from learning.

They tested how the most popular consciousness theories fit to the model under consideration. Their conclusion is that Information Integration Theory is the most suitable to demonstrate the consciousness of the model neurons.

Surely, it is a well-written and very interesting manuscript. However, I would like to ask the authors whether following the opposite approach would be more scientifically valid. To be more specific, all these theories have already demonstrated their utility and are of high scientific merit. So, maybe instead of investigating which one is most compatible to the newly introduced BNN model, why not investigating the similarities of this model with each one of those theories. Then, identifying some limitations of the proposed BNN model and suggesting opportunities for future work and updating the Kagan model.

In other words, it would be much more straightforward to consider the current theories as the golden standard and not the newly introduced model.

Finally, let me clarify that I do not oppose to the approach following by the authors but would like to read their arguments for their selection.

Another comment, deals with the Table. It would be useful if this Table would be cited repeatedly within the section 3 and serve as a connection point among the several IIT theory aspects and the Kagan model.

Specific Comments:

Within the Abstract section, the authors state that “We found that Information Integration Theory (IIT) is the only NCC that offers the possibility for a BNN to show consciousness”. I may agree with that argument, but the authors should justify it in terms of Φ value range.

They also state that creating BNNs with conscious capabilities may have physiological, philosophical, and ethical perspectives. What about real-world applications in medicine, gaming, transportation, or even military applications?

To sum up, my suggestion for the Abstract Section is to reformulate it a bit to explain the current consciousness theories and how they could be applied to biological neural networks. It would be also useful to answer several questions such as:

·        Why is it important to enable BNNs with consciousness?

·        Which is the type of study, brief methodological explanation.

Section 2

The authors state the following “In other words, a materialistic perspective on the easy problem of consciousness brings immediate rewards”. This argument is to favor a more practical approach on solving mind-body problems while not considering theoretical models of consciousness with philosophical background. I acknowledge, that this a more straightforward approach which can produce immediate results. On the other hand, I feel a bit anxious with bioengineering approaches lacking a theoretical model. It may not be a proper way of science and may lead to superficial approaches. Maybe the following reformulation of that sentence would be more useful: “we are adopting the current theoretical model of consciousness, while targeting on more practical applications.”

The following paragraph (lines 100-106) is also very abstract. Please identify the conditions and the causes for each one of the binary outcomes.

It is a bit range to reach the conclusion of the most useful theory (lines 119-122) prior to the description of those theories. Maybe a redesign of the manuscript’s structure is needed here.

Please correct they typo in line 131.

In line 132 I cannot understand what “value systems” mean. I assume that it has to do with evaluation of actions, but maybe a reformulation is needed here.

According to lines 151-154, it seems that GNW was not selected since it requires the existence of several network configurations (e.g. long memory structure) that are not yet available for implementing the neuronal system described by Kagan et al. So, this is a technological limitation and not a drawback of this theory in comparison with the IIT. Maybe future work would make it applicable for implementation in BNNs.

I am not sure I understand lines 189-190: “but it does not open the possibility for any alternative system to be conscious”. Shall the authors explain it a bit? According to my understanding, this theory involves the how a stimulus (visual) is processed and may lead to some actions. They state that only the cortex is involved in that processing. What happens in a threatening stimulus which may lead to the “fight or fly” reflex? In that case, we know that there is a strong involvement of the human amygdala (sub-cortical process).

There is the abbreviation of “P consciousness” in line 217. Since, it is the first time used in the manuscript, shall we also mention its full name? I assume that it is associated with phenomenology but am not sure.

There are some typos and other errors within this manuscript.

Please see my abovementioned comments.

Author Response

Response to Reviewer 2

General Comments:

The authors aim to investigate whether a recent work from “Kagan, B. J., Kitchen, A. C., Tran, N. T., Habibollahi, F., Khajehnejad, M., Parker, B. J., ... & Friston, K. J. (2022). In vitro neurons learn and exhibit sentience when embodied in a simulated game-world. Neuron110(23), 3952-3969.” Is equipped with some form of consciousness apart from learning.

They tested how the most popular consciousness theories fit to the model under consideration. Their conclusion is that Information Integration Theory is the most suitable to demonstrate the consciousness of the model neurons.

Surely, it is a well-written and very interesting manuscript. However, I would like to ask the authors whether following the opposite approach would be more scientifically valid. To be more specific, all these theories have already demonstrated their utility and are of high scientific merit. So, maybe instead of investigating which one is most compatible to the newly introduced BNN model, why not investigating the similarities of this model with each one of those theories. Then, identifying some limitations of the proposed BNN model and suggesting opportunities for future work and updating the Kagan model.

In other words, it would be much more straightforward to consider the current theories as the golden standard and not the newly introduced model.

Finally, let me clarify that I do not oppose to the approach following by the authors but would like to read their arguments for their selection.

Answer

The NCC theories aim to understand how neural processes relate to conscious experience. One of these theories is the Integrated Information Theory (IIT), which posits that consciousness arises from the integration of information in a system. When considering the applicability of IIT to explain consciousness in lower organisms, several factors make it a compelling choice.

Among IIT’s strengths, we can cite the following:

  • Predictive Power: IIT can make predictions about the level of consciousness in a system based on its neural connectivity and dynamics. This predictive aspect is particularly useful when studying lower organisms, where direct measurements of consciousness might be more challenging. By analyzing the neural structure and activity patterns, researchers can estimate the potential for consciousness in these organisms.
  • Quantitative Framework: IIT provides a quantitative measure called Phi (Φ) that quantifies the degree of information integration in a system. This allows for a systematic way to compare different systems and determine their level of consciousness. In the case of lower organisms, having a quantitative measure can help assess their conscious capacities more objectively.
  • Non-Anthropocentric Perspective: IIT doesn't rely on human-like cognitive abilities or complex neural structures. It can be applied to a wide range of systems, making it suitable for explaining consciousness in lower organisms that might have simpler neural networks.

From our understanding of the general comment, the reviewer is suggesting to overturn the main focus of the article by centering on the compatibility of each neural correlates of consciousness (NCC) theory with the newly developed BNN’s. We need to call attention here to the fact that our main interest was not the compatibility between NCC theories, but their predictive power. Our main question is which theory predicted the possibility of consciousness in a BNN, not which one is more compatible with its existence. We are trying to figure out what parts of these theories allow for the behavior the BNN shows. We clearly found out that none of the theories seem to even allow for any form of consciousness on a BNN due to their dependence on a thalamocortical model. And indeed, these theories were taken, as the reviewer indicates, as “golden standards” in the field of Neural Correlates of Consciousness. There is no denying of their heuristic value.

In addition, the article was requested as a collaboration for a new Special Issue entitled "Integrated Information Theory and Consciousness II.”  We may as well not depart very far from that subject by advocating for other theories.

Comment

  • Another comment, deals with the Table. It would be useful if this Table would be cited repeatedly within the section 3 and serve as a connection point among the several IIT theory aspects and the Kagan’s model.

Answer

We added the requested text citation in the MS.

Specific Comments:

Comment

Within the Abstract section, the authors state that “We found that Information Integration Theory (IIT) is the only NCC that offers the possibility for a BNN to show consciousness”. I may agree with that argument, but the authors should justify it in terms of Φ value range.

Answer

The justification has been added in the Abstract section.

In the main body of our paper, we do indeed provide a more comprehensive discussion of why IIT aligns with the potential for a BNN to exhibit consciousness, with a focus on the concept of Φ (Phi). We discuss how Φ, which quantifies the integrated information in a system, can serve as a measure of the degree of consciousness in a neural network. By emphasizing the role of Φ, we aim to provide a more robust argument for the unique position of IIT in this context.

Comment

They also state that creating BNNs with conscious capabilities may have physiological, philosophical, and ethical perspectives. What about real-world applications in medicine, gaming, transportation, or even military applications?

Answer

The reviewer is correct in this case, and mentions to those fields have been added to the text. However, these issues fall outside our present focus on theoretical analysis. We believe that real-world applications will be eventually instantiated, but it will be desirable, at least for once, to establish a philosophical and ethical background first, before those applications arrive. 

Comment

  • To sum up, my suggestion for the Abstract Section is to reformulate it a bit to explain the current consciousness theories and how they could be applied to biological neural networks. It would be also useful to answer several questions such as:

·        Why is it important to enable BNNs with consciousness?

·        Which is the type of study, brief methodological explanation.

Answer

The answer to these questions may show different facets depending on the point of view from where we depart. I’m not sure that gaining consciousness was the main intention in Kagan’s experiment, but simply the study of neural mechanisms associated to specific cognitive functions. Consciousness in these cases may happen as a side effect of the BNN implementation.

Regarding the request of reformulating the abstract to include the description of all the NCC theories, is physically impossible due to space limitation.

Section 2

Comment

The authors state the following “In other words, a materialistic perspective on the easy problem of consciousness brings immediate rewards”. This argument is to favor a more practical approach on solving mind-body problems while not considering theoretical models of consciousness with philosophical background. I acknowledge, that this a more straightforward approach which can produce immediate results. On the other hand, I feel a bit anxious with bioengineering approaches lacking a theoretical model. It may not be a proper way of science and may lead to superficial approaches. Maybe the following reformulation of that sentence would be more useful: “we are adopting the current theoretical model of consciousness, while targeting on more practical applications.”

Answer

  • We agree on principle. While we favor these kinds of experimental development, we also share the anxiety about future developments utilizing an indiscriminate  “brain in a box” approach. In order not to complicate matters more and make the meaning clear, we added some cautionary language to our support of BNN’s development. Both sides of the conflict mentioned by the reviewer have to be taken into consideration.

Comment

The following paragraph (lines 100-106) is also very abstract. Please identify the conditions and the causes for each one of the binary outcomes.

Answer

The passage has been rewritten.

Comment

It is a bit range (strange?) to reach the conclusion of the most useful theory (lines 119-122) prior to the description of those theories. Maybe a redesign of the manuscript’s structure is needed here.

Answer

The text has been re-arranged to avoid this confusion.

Comment

Please correct they typo in line 131.

Answer

“Evaluational” has been changed to “evaluative”

Comment

In line 132 I cannot understand what “value systems” mean. I assume that it has to do with evaluation of actions, but maybe a reformulation is needed here.

Answer

This falls outside the scope of our article. However,  Mashour et al (2020) indicates that “Baars’s global workspace involves processors related to the past (memory), present (sensory input, attention), and future (value systems, motor plans, verbal report)” There are no specific definitions of term in the text, but one can assume that it refers to the value of possible future actions in terms of punishment or reward.

Comment

According to lines 151-154, it seems that GNW was not selected since it requires the existence of several network configurations (e.g. long memory structure) that are not yet available for implementing the neuronal system described by Kagan et al. So, this is a technological limitation and not a drawback of this theory in comparison with the IIT. Maybe future work would make it applicable for implementation in BNNs.

Answer

The reviewer is correct in this regard and a comment to the effect has been added in the text.

Comment

I am not sure I understand lines 189-190: “but it does not open the possibility for any alternative system to be conscious”. Shall the authors explain it a bit? According to my understanding, this theory involves the how a stimulus (visual) is processed and may lead to some actions. They state that only the cortex is involved in that processing. What happens in a threatening stimulus which may lead to the “fight or fly” reflex? In that case, we know that there is a strong involvement of the human amygdala (sub-cortical process).

Answer

The reviewer is correct regarding fearful stimuli processed by the amygdala, but Lemma’s theory, again, focuses heavily on cortical processing, with no mention of subcortical structures. We feel that discussing these kinds of possible outcomes fall outside the scope of our MS.

Regarding explanations, the following paragraph in our mMS indicates the following “Our organoid instantiated in a dish demonstrates a direct connection between sensory and motor areas, showing an example of what Lamme calls fast feedforward sweep (FFS), but that is how far the similarities go. There is no higher-order system in which the original percept is processed at a higher order area. In addition, the central requirement of this theory, recurrent processing, is very limited in the internal connectivity of the organoid since there are no cortical layers nor complex cortical areas. Once again, this theory does not predict or explain Kagan et al.s results.”

Comment

There is the abbreviation of “P consciousness” in line 217. Since, it is the first time used in the manuscript, shall we also mention its full name? I assume that it is associated with phenomenology but am not sure.

Answer

The correction has been added to the text.

Round 2

Reviewer 2 Report

I would like to congratulate the authors for their work towards the improvement of this manuscript. Well done!